# SHINE: SHARING THE INVERSE ESTIMATE FROM THE FORWARD PASS FOR BI-LEVEL OPTIMIZATION AND IMPLICIT MODELS

**Zaccharie Ramzi**
CEA (Neurospin & Cosmostat)
Inria (Parietal)
Gif-sur-Yvette, France
`zaccharie.ramzi@inria.fr`

**Florian Mannel**
University of Graz
Graz, Austria

**Shaojie Bai**
Carnegie Mellon University
Pittsburgh, USA

**Jean-Luc Starck**
AIM, CEA, CNRS
Gif-sur-Yvette, France

**Philippe Ciuciu**
CEA (Neurospin), Inria (Parietal)
Gif-sur-Yvette, France

**Thomas Moreau**
Inria (Parietal)
Gif-sur-Yvette, France

## ABSTRACT

In recent years, implicit deep learning has emerged as a method to increase the effective depth of deep neural networks. While their training is memory-efficient, they are still significantly slower to train than their explicit counterparts. In Deep Equilibrium Models (DEQs), the training is performed as a bi-level problem, and its computational complexity is partially driven by the iterative inversion of a huge Jacobian matrix. In this paper, we propose a novel strategy to tackle this computational bottleneck from which many bi-level problems suffer. The main idea is to use the quasi-Newton matrices from the forward pass to efficiently approximate the inverse Jacobian matrix in the direction needed for the gradient computation. We provide a theorem that motivates using our method with the original forward algorithms. In addition, by modifying these forward algorithms, we further provide theoretical guarantees that our method asymptotically estimates the true implicit gradient. We empirically study this approach and the recent Jacobian-Free method in different settings, ranging from hyperparameter optimization to large Multiscale DEQs (MDEQs) applied to CIFAR and ImageNet. Both methods reduce significantly the computational cost of the backward pass. While SHINE has a clear advantage on hyperparameter optimization problems, both methods attain similar computational performances for larger scale problems such as MDEQs at the cost of a limited performance drop compared to the original models.

## 1 INTRODUCTION

Implicit deep learning models such as Neural ODEs (Chen et al., 2018), OptNets (Amos and Zico Kolter, 2017) or Deep Equilibrium models (DEQs) (Bai et al., 2019; 2020) have recently emerged as a way to train deep models with infinite effective depth without the associated memory cost. Indeed, while it has been observed that the performance of deep learning models increases with their depth (Telgarsky, 2016), an increase in depth also translates into an increase in the memory footprint required for training, which is hardware-constrained. While other works such as invertible neural networks (Gomez et al., 2017; Sander et al., 2021) or gradient checkpointing (Chen et al., 2016) also tackle this issue, implicit models bear an $\mathcal{O}(1)$ memory cost and with constraints on the architecture that are usually not detrimental to the performance (Bai et al., 2019). These models have been successfully applied to large-scale tasks such as language modeling (Bai et al., 2019), computer vision (Bai et al., 2020) and inverse problems (Gilton et al., 2021; Heaton et al., 2021).

In general, the formulation of DEQs can be cast as a bi-level problem of the following form:

$$\arg\min_{\theta} \mathcal{L}(z^{\star}) \quad \text{subject to} \quad g_{\theta}(z^{\star}) = 0. \tag{1}$$

We will refer to the root finding problem $g_{\theta}(z^{\star}) = 0$ as the *inner problem*, and call its resolution the *forward pass*. On the other hand, we will refer to $\arg\min_{\theta} \mathcal{L}(z^{\star})$ as the *outer problem*, and

call the computation of the gradient of $\mathcal{L}(z^\star)$ w.r.t. $\theta$ the *backward pass*. The core idea for DEQs is that their output $z^\star$ is expressed as a fixed point of a parametric function $f_\theta$ from $\mathbb{R}^d$ to $\mathbb{R}^d$, i.e., $g_\theta(z^\star) = z^\star - f_\theta(z^\star) = 0$.[1] This model is said to have infinitely many weight-tied layers as $z^\star$ can be obtained by successively applying the layer $f_\theta$ infinitely many times, provided $f_\theta$ is contractive. In practice, DEQs' forward pass is not computed by applying successively the function but usually relies on quasi-Newton (qN) algorithms, such as Broyden's method (Broyden, 1965), which approximates efficiently the Jacobian matrix $\frac{\partial g_\theta}{\partial z}$ and its inverse for root-finding.

To compute DEQs' gradient efficiently and avoid high memory cost, one does not rely on back-propagation but uses the implicit function theorem (Krantz and Parks, 2013) which gives an analytical expression of the Jacobian of $z^\star$ with respect to $\theta$, $\frac{\partial z^\star}{\partial \theta}$. While this method is memory efficient, it requires the computation of matrix-vector products involving the inverse of a large Jacobian matrix, which is computationally demanding. To make this computation tractable, one needs to rely on an iterative algorithm based on vector-Jacobian products, which renders the training particularly slow, as highlighted by the original authors (Bai et al., 2020) (see also the break down of the computational effort in Section E.4).

Moreover, the formulation (1) allows us to also consider general bi-level problems such as hyper-parameter optimization under the same framework. For instance, hyperparameter optimization for Logistic Regression (LR) can be written as

$$\min_\theta \mathcal{L}_{\text{val}}(z^*) \quad \text{subject to} \quad z^* = \min_z r_\theta(z) \triangleq \mathcal{L}_{\text{train}}(z) + \theta\|z\|_2^2, \tag{2}$$

where $\mathcal{L}_{\text{train}}$ and $\mathcal{L}_{\text{val}}$ correspond to the training and validation losses from the LR problem (Pedregosa, 2016). Here, $z$ corresponds to the weights of the LR model while $\theta$ is the regularisation parameter. As the training loss is smooth and convex, the inner problem can be written as in (1) with $g_\theta = \nabla_z r_\theta$ to fit (1). Similarly to DEQ, the inner problem is often solved using qN methods, which approximate the inverse of the Hessian in the direction of the steps, such as the LBFGS algorithm (Liu and Nocedal, 1989), and the gradient computation suffers from the same drawback as it is also obtained using the implicit function theorem. Lorraine et al. (2020) review the different hypergradient approximations for bi-level optimization and evaluate them on multiple tasks.

With the increasing popularity of DEQs and the ubiquity of bi-level problems in machine learning, a core question is how to reduce the computational cost of the resolution of (1). This would make these methods more accessible for practitioners and reduce the associated energy cost. In this work, we propose to exploit the estimates of the (inverse of the) Jacobian/Hessian produced by qN methods in the hypergradient computation. Moreover, we also propose extra updates of the qN matrices which maintain the approximation property in the direction of the steps, and ensure that the inverse Jacobian is approximated in an additional direction. In effect, we can compute the gradient using the inverse of the final qN matrix instead of an iterative algorithm to invert the Jacobian in the gradient's direction, while stressing that the inverse of a qN matrix, and thus the multiplication with it, can be computed very efficiently.

We emphasize that the goal of this paper is neither to improve the algorithms used to compute $z^\star$, nor is it to demonstrate how to perform the inversion of a matrix in a certain direction as a stand-alone task. Rather, we are describing an approach that combines the resolution of the inner problem with the computation of the hypergradient to accelerate the overall process. Our work is the first to consider modifying the inner problem resolution in order to account for the bi-level structure of the optimization The idea to use additional updates of the qN matrices to ensure additional approximation properties is not new, and it is also known that a full matrix inversion can be accomplished in this way. For instance, Gower and Richtárik (2017) used sketching to design appropriate extra secant conditions in order to obtain guarantees of uniform convergence towards the inverse of the Jacobian. The novelty in our work is that we integrate additional update to yield the inverse in a specific direction, which is substantially cheaper than computing the inverse. A concurrent work by Fung et al. (2021) is also concerned with the acceleration of DEQs' training, where the inverse Jacobian is approximated with the identity. Under strong contractivity and conditioning assumptions, it is proven that the resulting approximation is a descent direction and the authors show good empirical performances for small scale problems.

The contributions of our paper are the following:

---

[1]Here, we do not explicitly write the dependence of $f_\theta$ on the input $x$ of the DEQ, usually referred to as the injection.

- We introduce a new method to greatly accelerate the backward pass of DEQs (and generally, the differentiation of bi-level problems) using qN matrices that are available as a by-product of the forward computations. We call this method **SHINE** (**SH**aring the **IN**verse **E**stimate).

- We enhance this method by incorporating knowledge from the outer problem into the inner problem resolution. This allows us to provide strong theoretical guarantees for this approach in various settings.

- We additionally showcase its use in hyperparameter optimization. Here, we demonstrate that it provides a gain in computation time compared to state-of-the-art methods.

- We test it for DEQs for the classification task on two datasets, CIFAR and ImageNet. Here, we show that it decreases the training time while remaining competitive in terms of performance.

- We extend the empirical evaluation of the Jacobian-Free method to large scale multiscale DEQs and show that it performs well in this setting. We also show that it is not suitable for more general bi-level problems.

- We propose and evaluate a natural refinement strategy for approximate Jacobian inversion methods (both SHINE and Jacobian-Free) that allows a trade-off between computational cost and performances.

## 2 HYPERGRADIENT OPTIMIZATION WITH APPROXIMATE JACOBIAN INVERSE

### 2.1 SHINE: HYPERGRADIENT DESCENT WITH APPROXIMATE JACOBIAN INVERSE

**Hypergradient Optimization**   Hypergradient optimization is a first-order method used to solve (1). We recall that in the case of smooth convex optimization, $\frac{\partial g_\theta}{\partial z}$ is the Hessian of the inner optimization problem, while for deep equilibrium models, it is the Jacobian of the root equation. In the rest of this paper, with a slight abuse of notation, we will refer to both these matrices with $J_{g_\theta}$ whenever the results can be applied to both contexts. To enable Hypergradient Optimization, i.e. gradient descent on $\mathcal{L}$ with respect to $\theta$, Bai et al. (2019, Theorem 1) show the following theorem, which is based on implicit differentiation (Krantz and Parks, 2013):

**Theorem 1** (Hypergradient (Bai et al., 2019; Krantz and Parks, 2013)). *Let $\theta \in \mathbb{R}^p$ be a set of parameters, let $\mathcal{L} : \mathbb{R}^d \to \mathbb{R}$ be a loss function and $g_\theta : \mathbb{R}^d \to \mathbb{R}^d$ be a root-defining function. Let $z^\star \in \mathbb{R}^d$ such that $g_\theta(z^\star) = 0$ and $J_{g_\theta}(z^\star)$ is invertible, then the gradient of the loss $\mathcal{L}$ wrt. $\theta$, called Hypergradient, is given by*

$$\frac{\partial \mathcal{L}}{\partial \theta}\bigg|_{z^\star} = \nabla_z \mathcal{L}(z^\star)^\top J_{g_\theta}(z^\star)^{-1} \frac{\partial g_\theta}{\partial \theta}\bigg|_{z^\star}. \quad (3)$$

In practice, we use an algorithm to approximate $z^\star$, and Theorem 1 gives a plug-in formula for

---

**Algorithm 1:** qN method to solve $g_\theta(z^\star) = 0$

**Result:** Root $z^\star$, qN matrix $B$
$b = \texttt{true}$ if using Broyden's method,
  $b = \texttt{false}$ if using BFGS
$n = 0$, $z_0 = 0$, $B_0 = I$
**while** *not converged* **do**
  $p_n = -B_n^{-1} g_\theta(z_n)$, $z_{n+1} = z_n + \alpha_n p_n$
    `// `$\alpha_n$` can be 1 or determined`
    `by line-search`
  $y_n = g_\theta(z_{n+1}) - g_\theta(z_n)$
  $s_n = z_{n+1} - z_n$
  **if** $b$ **then**
    $B_{n+1} = \underset{X:\, Xs_n = y_n}{\arg\min} \|X - B_n\|_F$
  **else**
    $B_{n+1} =$
      $\underset{X:\, X = X^T \wedge Xs_n = y_n}{\arg\min} \|X^{-1} - B_n^{-1}\|$
    `// The norm used in BFGS`
    `is a weighted Frobenius`
    `norm`
  **end**
  $n \leftarrow n + 1$
**end**
$z^\star = z_n$, $B = B_n$

---

the backward pass. Note that this formula is independent of the algorithm chosen to compute $z^\star$. Moreover, as opposed to explicit networks, we do not need to store intermediate activations, resulting in the aforementioned training time memory gain for DEQs. Once $z^\star$ has been obtained, one of the major bottlenecks in the computation of the Hypergradient is the inversion of $J_{g_\theta}(z^\star)$ in the directions $\frac{\partial g_\theta}{\partial \theta}\big|_{z^\star}$ or $\nabla_z \mathcal{L}(z^\star)$.

**Quasi-Newton methods**   In practice, the forward pass is often carried out with qN methods. For instance, in the case of bi-level optimization for Logistic Regression, Pedregosa (2016) used L-BFGS (Liu and Nocedal, 1989), while for Deep Equilibrium Models, Bai et al. (2019) used Broyden's

method (Broyden, 1965), later adapted to the multi-scale case in a limited-memory version (Bai et al., 2020).

These quasi-Newton methods were first inspired by Newton's method, which finds the root of $g_\theta$ via the recurrent Jacobian-based updates $z_{n+1} = z_n - J_{g_\theta}(z_n)^{-1}g_\theta(z_n)$. Specifically, they replace the Jacobian $J_{g_\theta}(z_n)$ by an approximation $B_n$ that is based on available values of the iterates $z_n$ and $g_\theta$ rather than its derivative. These $B_n$, called qN matrices, are defined recursively via an optimization problem with constraints called secant conditions. Solving this problem leads to expressing $B_n$ as a rank-one or rank-two update of $B_{n-1}$, so that $B_n$ is the sum of the initial guess $B_0$ (in our settings, the identity) and $n$ low-rank matrices (less than $n$ in limited memory settings). This low rank structure allows efficient multiplication by $B_n$ and $B_n^{-1}$. We now explain how the use of qN methods as inner solver can be exploited to resolve this computational bottleneck.

**SHINE**   Roughly speaking, our proposition is to use $B^{-1} = \lim_{n\to\infty} B_n^{-1}$ as a replacement for $J_{g_\theta}(z^\star)^{-1}$ in (3), i.e. to share the inverse estimate between the forward and the backward passes. This gives the approximate Hypergradient

$$p_\theta = \nabla_z \mathcal{L}(z^\star) B^{-1} \frac{\partial g_\theta}{\partial \theta}\Big|_{z^\star}. \tag{4}$$

In practice we will consider the nonasymptotical direction $p_\theta^{(n)} = \nabla_z \mathcal{L}(z_n) B_n^{-1} \frac{\partial g_\theta}{\partial \theta}\Big|_{z_n}$. Thanks to the Sherman-Morrison formula (Sherman and Morrison, 1950), the inversion of $B_n$ can be done very efficiently (using scalar products) compared to the iterative methods needed to invert the true Jacobian $J_{g_\theta}(z^\star)$. In turn, this significantly reduces the computational cost of the Hypergradient computation.

**Relationship to the Jacobian-Free method**   Because $B_0 = I$ in our setting, we may regard $B$ as an identity matrix perturbed by a few rank-one updates. In the directions that are used for updates, $B$ is going to be different from the identity, and hopefully closer to the true Jacobian in those directions. However, in all orthogonal directions we fall exactly into the setting of the Jacobian-Free method introduced by Fung et al. (2021). In that work, $J_{g_\theta}(z^\star)^{-1}$ is approximated by $I$, and the authors highlight that this is equivalent to using a preconditioner on the gradient. Under strong assumptions on $g_\theta$ they show that this preconditioned gradient is still a descent direction.

**Transition to the exact Jacobian Inverse.**   The approximate gradient $p_\theta^{(n)}$ can also be used as the initialization of an iterative algorithm for inverting $J_{g_\theta}(z^\star)$ in the direction $\nabla_z \mathcal{L}(z^\star)$. With a good initialization, faster convergence can be expected. Moreover, if the iterative algorithm is also a qN method, which is the case in practice in the DEQ implementation, we can use the qN matrix $B$ from the forward pass to initialize the qN matrix of this algorithm. We refer to this strategy as the *refine strategy*. Because the refine strategy is essentially a smart initialization scheme, it recovers all the theoretical guarantees of the original method (Bai et al., 2019; 2020; Pedregosa, 2016).

## 2.2   CONVERGENCE TO THE TRUE GRADIENT

To further justify and formalize the idea of SHINE, we show that the direction $p_\theta^{(n)}$ converges to the Hypergradient $\frac{\partial \mathcal{L}}{\partial \theta}\Big|_{z^\star}$. We now collect the assumptions that will be used for this purpose.

**Assumption 1** (Uniform Linear Independence (ULI) (Li et al., 1998)). *There exist a positive constant $\rho > 0$ and natural numbers $n_0 \geq 0$ and $m \geq d$ with the following property: For any $n \geq n_0$ we can find indices $n \leq n_1 \leq \ldots \leq n_d \leq n + m$ such that, for $p_n$ defined in Algorithm 1, the smallest singular value of the $d \times d$ matrix*

$$\left( \frac{p_{n_1}}{\|p_{n_1}\|}, \ \frac{p_{n_2}}{\|p_{n_2}\|}, \ \ldots, \ \frac{p_{n_d}}{\|p_{n_d}\|} \right)$$

*is no smaller than $\rho$.*

**Assumption 2** (Smoothness and convergence to the fixed point). *(i) $\sum_{n=0}^\infty \|z_n - z^\star\| < \infty$ for some $z^\star$ with $g_\theta(z^\star) = 0$; (ii) $g_\theta$ is $C^1$, $J_{g_\theta}$ is Lipschitz continuous near $z^\star$, and $J_{g_\theta}(z^\star)$ is invertible; (iii) $\nabla_z \mathcal{L}$ is continuous, and $\forall \theta$, $\frac{\partial g_\theta}{\partial \theta}$ is continuous.*

**Remark.** *The Assumption 2 (i) implies $\lim_{n\to\infty} z_n = z^\star$. The existence of the Jacobian and its inverse are assumptions that are already made in the regular DEQ setting just to train the model.*

**Theorem 2** (Convergence of SHINE to the Hypergradient using ULI). *Let us denote $p_\theta^{(n)}$, the SHINE direction for iterate $n$ in Algorithm 1 with $b =$ true. Under Assumptions 1 and 2, for a given parameter $\theta$, $(z_n)$ converges q-superlinearly to $z^\star$ and*

$$\lim_{n \to \infty} p_\theta^{(n)} = \frac{\partial \mathcal{L}}{\partial \theta}\Big|_{z^\star}.$$

*Proof.* From More and Trangenstein (1976, Theorem 5.7) we obtain that $\lim_{n \to \infty} B_n = J_{g_\theta}(z^\star)$. We can then conclude using the continuity of the inversion operator on the space of invertible matrices and of the right and left matrix vector multiplications. A complete proof is given in Section B.1. □

Theorem 2 establishes convergence of the SHINE direction to the true Hypergradient, but relies on Assumption 1 (ULI). While ULI is often used to prove convergence results for qN matrices, e.g. in (Conn et al., 1991; Li et al., 1998; Nocedal and Wright, 2006), it is a strong assumption whose satisfaction in practice is debatable, cf., e.g., (Fayez Khalfan et al., 1993). For Broyden's method, ULI is violated in all numerical experiments in (Mannel, 2020; 2021a;b), and those works also prove that ULI is necessarily violated in certain settings (but the setting of this work is not covered). In the following we therefore derive results that do not involve ULI.

### 2.3 OUTER PROBLEM AWARENESS

The ULI assumption guarantees convergence of $B_n^{-1}$ to $J_{g_\theta}(z^\star)^{-1}$. However, (3) only requires the multiplication of $J_{g_\theta}(z^\star)^{-1}$ with $\frac{\partial g_\theta}{\partial \theta}|_{z^\star}$ from the right and $\nabla_z \mathcal{L}(z^\star)$ from the left.

**BFGS with OPA** In order to strengthen Theorem 2, let us consider the setting of bi-level optimization with a single regularizing hyperparameter $\theta$. There, the partial derivative $\frac{\partial g_\theta}{\partial \theta}|_{z^\star}$ is a $d$-dimensional vector and it is possible to compute its approximation $\frac{\partial g_\theta}{\partial \theta}|_{z_n}$ at a reasonable cost. We propose to incorporate additional updates of the quasi-Newton matrix $B_n$ into Algorithm 1 that improve the approximation quality of $B_n^{-1}$ in the direction $\frac{\partial g_\theta}{\partial \theta}|_{z_n}$ (thus asymptotically in the direction $\frac{\partial g_\theta}{\partial \theta}|_{z^\star}$). Given a current iterate pair $(z_n, B_n)$, these additional updates only change $B_n$, but not $z_n$. We will demonstrate that a suitable update direction $e_n \in \mathbb{R}^d$ is given by

$$e_n = t_n B_n^{-1} \frac{\partial g_\theta}{\partial \theta}\Big|_{z_n}, \tag{5}$$

where $(t_n) \subset [0, \infty)$ satisfies $\sum_n t_n < \infty$. This update direction will be used to create an extra secant condition $X^{-1}(g_\theta(z_n + e_n) - g_\theta(z_n)) = e_n$ for the additional update of $B_n$. Since this extra update is based on the outer problem, we refer to this technique as Outer-Problem Awareness (OPA). The complete pseudo code of the OPA method in the LBFGS algorithm (Liu and Nocedal, 1989) is given in Appendix A.
We now prove that if extra updates are applied at a fixed frequency, then fast (q-superlinear) convergence of $(z_n)$ to $z^\star$ is retained, while convergence of the SHINE direction to the true Hypergradient is also ensured. To show this, we use the following assumption.

**Assumption 3** (Assumptions for BFGS). *Let $g_\theta(z) = \nabla_z r_\theta(z)$ for some $C^2$ function $r_\theta : \mathbb{R}^d \to \mathbb{R}$. Consider Algorithm 1 with $b =$ false. We assume some regularity on $r$ and that an appropriate line search is used. An extended version of this assumption is given in Section B.2 (Assumption 5).*

**Theorem 3** (Convergence of SHINE to the Hypergradient for BFGS with OPA). *Let us consider $p_\theta^{(n)}$, the SHINE direction for iterate $n$ in Algorithm 1 that is enriched by extra updates in the direction $e_n$ defined in (5). Under Assumptions 2 (ii-iii) and 3, for a given parameter $\theta$, we have the following: Algorithm 1, for any symmetric and positive definite matrix $B_0$, generates a sequence $(z_n)$ that converges q-superlinearly to $z^\star$, and there holds*

$$\lim_{n \to \infty} p_\theta^{(n)} = \frac{\partial \mathcal{L}}{\partial \theta}\Big|_{z^\star}. \tag{6}$$

*Proof.* It follows from known results that the extra updates do not destroy the q-superlinear convergence of $(z_n)$. The proof of (6) relies firstly on the fact that by continuity of the derivative of $g_\theta$, we have $\lim_{n \to \infty} \frac{\partial g_\theta}{\partial \theta}|_{z_n} = \frac{\partial g_\theta}{\partial \theta}|_{z^\star}$. Due to the extra updates we can show convergence of the qN matrices to the true Hessian in the direction of the extra steps $e_n$, from which (6) follows. A full proof is provided in Section B.2. □

**Remark.** *Theorem 3 also holds without line searches (i.e., $\alpha_n = 1$ for all $n$) and any $C^2$ function $r_\theta$ (such that $g_\theta(z) = \nabla_z r_\theta(z)$) with locally Lipschitz continuous Hessian if $z_0$ is close enough to some $z^\star$ with $\nabla_z r_\theta(z^\star) = 0$ and $\nabla_{zz}^2 r_\theta(z^\star)$ positive definite.*

We note that Theorem 3 guarantees fast convergence of the iterates $(z_n)$ and that $z_0$ does not have to be close to $z^\star$ for that guarantee. Also, there is no restriction on $B_0$ other than being symmetric and positive definite (which is satisfied for our choice $B_0 = I$). Finally, Theorem 3 does not rely on ULI. From a practical standpoint we thus regard Theorem 3 as a much stronger result than Theorem 2.

**Adjoint Broyden with OPA**    It is not practical to use the partial derivative $\frac{\partial g_\theta}{\partial \theta}$ in the DEQ setting because it is a huge Jacobian that we do not have access to in practice. In order to still leverage the core idea of OPA, we propose to use extra updates that ensure that $B_n^{-1}$ approximates $J_{g_\theta}(z^\star)^{-1}$ in the direction $\nabla_z \mathcal{L}(z^\star)$ applied by left-multiplication, as required by (3). An appropriate secant condition is given by

$$v_n^T B_{n+1} = v_n^T J_{g_\theta}(z_{n+1}), \tag{7}$$

where

$$v_n^T = \nabla_z \mathcal{L}(z_n) B_n^{-1}. \tag{8}$$

To incorporate the secant condition (7), we use the Adjoint Broyden's method (Schlenkrich et al., 2010), a qN method relying on the efficient vector-Jacobian multiplication by $J_{g_\theta}$ using auto-differentiation tools. To prove convergence of the SHINE direction for this method, we need the following assumption.

**Assumption 4** (Uniform boundedness of the inverse qN matrices). *The sequence $(B_n)$ generated by Algorithm 1 satisfies*

$$\sup_{n \in \mathbb{N}} \|B_n^{-1}\| < \infty.$$

**Remark.** *Convergence results for quasi-Newton methods usually include showing that Assumption 4 holds, cf. Broyden et al. (1973, Theorem 3.2) for Broyden's method and the BFGS method, respectively, Schlenkrich et al. (2010, Theorem 1) for the Adjoint Broyden's method. It can also be proved that Assumption 4 holds for globalized variants of these methods, e.g., for the line-search globalizations of Broyden's method proposed by Li and Fukushima (2000). We point out that Assumption 1 entails $\lim B_n = J_{g_\theta}(z^\star)$ and thus $\lim B_n^{-1} = J_{g_\theta}(z^\star)^{-1}$, so it is clearly stronger than Assumption 4.*

**Theorem 4** (Convergence of SHINE to the Hypergradient for Adjoint Broyden with OPA). *Let us consider $p_\theta^{(n)}$, the SHINE direction for iterate $n$ in Algorithm 1 with the Adjoint Broyden secant condition (7) and extra update in the direction $v_n$ defined in (8). Under Assumptions 2 and 4, for a given parameter $\theta$, we have q-superlinear convergence of $(z_n)$ to $z^\star$ and*

$$\lim_{n \to \infty} p_\theta^{(n)} = \left.\frac{\partial \mathcal{L}}{\partial \theta}\right|_{z^\star}.$$

*Proof.* The q-superlinear convergence of $(z_n)$ follows from Schlenkrich et al. (2010, Theorem 2). To establish convergence of the SHINE direction, we proceed in three steps. First, it is shown that for $\nabla_z \mathcal{L}(z^\star) = 0$ the claim holds due to continuity and Assumption 4. Then $\nabla_z \mathcal{L}(z^\star) \neq 0$ is considered and it is proved that the desired convergence holds on the subsequence that corresponds to the additional updates. Lastly, this result is transferred to the entire sequence by involving the fixed frequency of the additional updates. The complete proof is provided in Section B.3.  □

Using the Adjoint Broyden's method comes at a computational cost. Indeed, because we now rely on $J_{g_\theta}$, we have to store the activations of $g_\theta(z)$ (which has a computational cost in addition to a memory cost), but also perform the vector-Jacobian product in addition to the function evaluation.

## 3   RESULTS

We test our method in 3 different setups and compare it to the original iterative inversion and its closest competitor, the Jacobian-Free method (Fung et al., 2021). We draw the reader's attention to the fact that although the Jacobian-Free method (Fung et al., 2021) is used outside the assumptions needed to have theoretical guarantees[2] of descent, it still performs relatively well in the Deep Equilibrium setting. The same is true for SHINE: While the ULI assumption is not met (and we are in practice far from the fixed point convergence), it performs well in practice.

---

[2]See the results on contractivity in Section E.3.

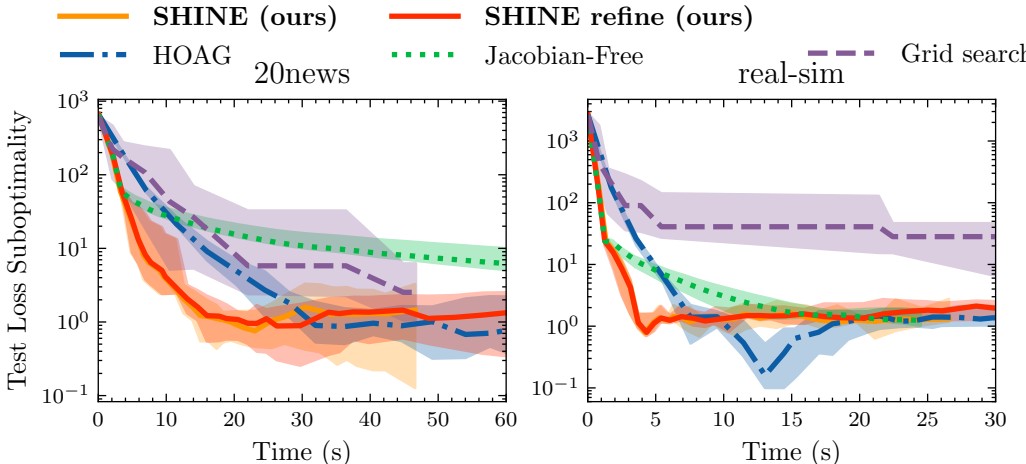

Figure 1: **Bi-level optimization:** Convergence of held-out test loss for different hyperparameter optimization methods on the $\ell_2$-regularized logistic regression problem for the 2 datasets (20news (Lang, 1995) and real-sim (lib)) SHINE achieves the best performances for both problems while the Jacobian-Free method is much slower, in particular on 20news. Note that the kink for HOAG on real-sim does not mean it is better as the optimization stops once the validation loss has converged and not the test one. The typical loss order of magnitude is $10^2$. An extended figure with more methods is provided in Section E.1.

**Implementations.** All the bi-level optimization experiments were done using the HOAG code (Pedregosa, 2016)[3], which is based on the Python scientific ecosystem (Harris et al., 2020; Pedregosa et al., 2011; Virtanen et al., 2020). Deep Equilibrium experiments were done using the PyTorch (Paszke et al., 2019) code for Multiscale DEQ (Bai et al., 2020)[4], which was distributed under the MIT license. Plots were done using Matplotlib (Hunter, 2007), with Science Plots style (Garrett and Peng, 2021). DEQ trainings were done in a publicly funded HPC, on nodes with 4 V100 GPUs. In practice, we never reach convergence of $(z_n)$, hence the approximate gradient might be far from the true gradient. To improve the approximation quality, we now propose a variant of our method.

**Fallback in the case of wrong inversion.** Empirically, we noticed that using $B$ can sometimes produce bad approximations, although with very low probability. We propose to detect this with by monitoring a telltale sign based on the norm of the approximation, as we verified on several examples that cases with a huge norm compared to the correct inversion also had a very bad correlation with the correct inversion. In these cases, we can simply fallback onto another inversion method. For the Deep Equilibrium experiments, when the norm of the inversion using SHINE is 1.3 times above the norm of the inversion using the Jacobian-Free method (which is available at no extra computational cost), we use the Jacobian-Free inversion. We refer to this strategy as the *fallback strategy*.

### 3.1 BI-LEVEL OPTIMIZATION – HYPERPARAMETER OPTIMIZATION IN LOGISTIC REGRESSION

We first test SHINE in the simple setting of bi-level optimization for $\ell_2$-regularized LR, using the code from Pedregosa (2016) and the same datasets. Convergence on unseen data is illustrated in Figure 1.[5] An acceptable level of performance is reached twice faster for the SHINE method compared to any other competitor. Another finding is that the refine strategy does not provide a definitive improvement over the vanilla version of SHINE. In order to verify that the performance gain of SHINE is not simply driven by truncated inversion, we also run HOAG with limited number of inversion iteration and showed that this degrades its performances (see HOAG limited backward in Section E.1).
We also tested our implementation of OPA on the 20news dataset and present the results in Figure 2. In order to get a fair comparison, we implemented both SHINE, SHINE-OPA and HOAG using the same

---

[3]https://github.com/fabianp/hoag
[4]https://github.com/locuslab/mdeq
[5]To facilitate the reader's understanding of the figures, we plot the empirical suboptimality, but we do remind them that there is no guarantee of convergence on held-out test data ; the kink present in the case of the real-sim dataset is an example of that.

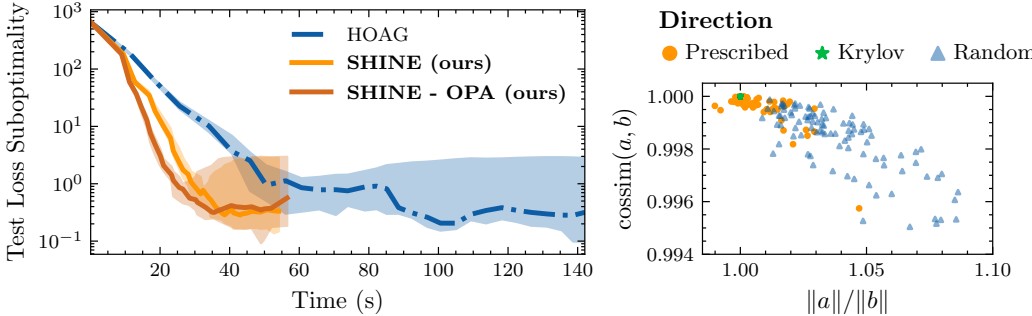

Figure 2: **Bi-level optimization with OPA:** (*left*) Convergence of different hyperparameter optimization methods on the $\ell_2$-regularized LR problem for the 20news dataset (Lang, 1995) on held-out test data. SHINE with OPA achieves similar performance as SHINE without OPA but with better convergence guarantees. (*right*) Evaluation of the inversion quality in direction $v$ using OPA $b = B_n^{-1}v$ compared to the exact inverse $a = J_{g_\theta}(z^\star)^{-1}v$ for 3 different directions: the prescribed direction, the Krylov direction and a random direction. The points represent the cosine similarity between $a$ and $b$ as a function of the ratio of their norm and the closer to $(1, 1)$ the better. The inverse in the prescribed direction is better than in random direction.

full Python code instead of relying on the original code which relied on the Fortran implementation of L-BFGS from (Virtanen et al., 2020). While SHINE with OPA does not outperform the vanilla SHINE, it reaches similar performances, outperforming HOAG, and comes with strong theoretical grounding. Additional results on hyperparameter optimization for the regularized nonlinear least squares problem are available in Section E.2.

We also showed on a smaller dataset, the breast cancer dataset (Dua and Graff, 2017), that OPA indeed ensures a better approximation of the inverse in the prescribed direction. For a given split of the data, we compared the quality of the approximation of the inversion in three different directions: a prescribed direction chosen randomly but used for the OPA update, the Krylov direction $\frac{\partial g_\theta}{\partial z}\Big|_{z^\star}(z_n - z_{n-1})$ and a random direction not used in the qN algorithm. The results for 100 runs with different random seeds are depicted in Figure 2, where we can observe that OPA indeed ensures a better inversion in the prescribed direction compared to a random direction. We also notice that a poor direction for the inversion seems correlated with a small magnitude.

## 3.2 DEEP EQUILIBRIUM MODELS

Next, we tested SHINE on the more challenging DEQ setup. Two experiments illustrate the performance of SHINE on the image classification task on two datasets. For both datasets, we used the same model configuration as in the original Multiscale DEQ paper (Bai et al., 2020) and did not fine tune any hyperparameter. For the different DEQ training methods, models for a given seed share the same unrolled-pretraining steps. We do not include OPA in the DEQ results because while the gradients are well correlated with the true ones (see Figure E.3), we observe a sharp initial performance drop that reduces its performance on Imagenet. We provide partial results in Section E.5.

**CIFAR-10.** The first dataset is CIFAR-10 (Krizhevsky, 2009) which features 60,000 $32{\times}32$ images representing 10 classes. For this dataset, the size of the multi-scale fixed point is $d = 50$k. We train the models for five different random seeds.

The results in Figure 3 show that for the vanilla version, SHINE slightly outperforms the Jacobian-Free method (Fung et al., 2021). Additionally, our results suggest that SHINE (in its vanilla version) is able to reduce the time taken for the backward pass almost 10-fold compared to the original method while retaining a competitive performance (on par with Res-Net-18 (He et al., 2016) at 92.9%). Finally, we do highlight that the Jacobian-Free method (Fung et al., 2021) is able to perform well outside the scope of its theoretical assumptions, albeit with slightly worse performance than SHINE. We conjecture that the batched stochastic gradient descent helps accelerated methods by averaging out the errors made in the approximation.

**ImageNet.** The second dataset is the ImageNet dataset (Deng et al., 2009) which features 1.2 million images cropped to $224{\times}224$, representing 1000 classes. This dataset is recognized as a large-scale computer vision problem and the dimension of the fixed point to find is $d = 190$k.

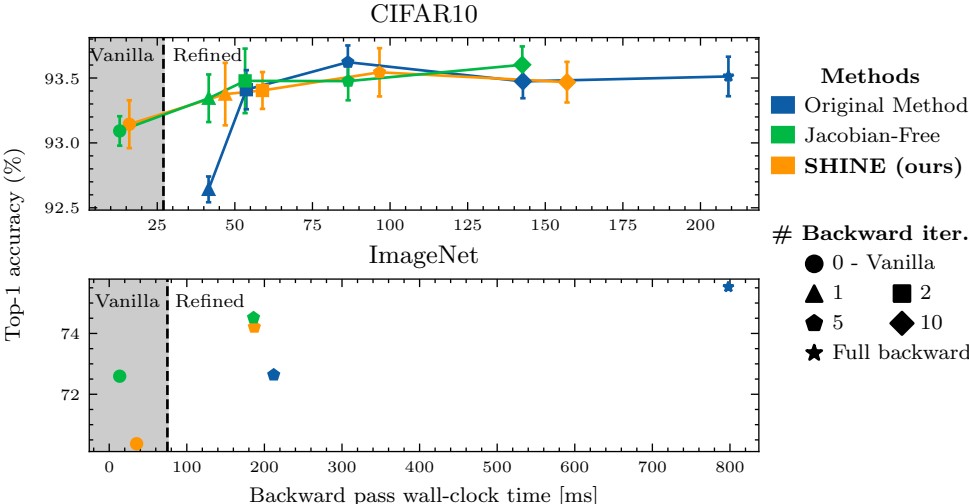

Figure 3: **DEQ:** Top-1 accuracy function of backward pass runtime for the different methods considered to train DEQs, on CIFAR (Krizhevsky, 2009) and ImageNet (Deng et al., 2009). The original DEQ training method corresponds to the Full backward pass points and the vanilla SHINE and Jacobian-Free methods correspond to direct use of the inverse approximation without further refinement. The other points correspond to further refinements of the different methods with different number of iterations used to invert $J_{g_\theta}(z^\star)$ in the direction of $\nabla_z \mathcal{L}(z^\star)$. This highlights the trade-off between computations and performances driving the refinement choice.

For this challenging task, we noticed that the vanilla version of SHINE was suffering a big drop just after the transition from unrolled pre-training to actual equilibrium training. To remedy partly this problem, we introduced the fallback to Jacobian-Free inversion. The results for a single random seed presented in Figure 3 for the ImageNet dataset are given for SHINE with fallback. The fallback is barely used : in 1000 batches of size 32, only 2 samples used fallback, a proportion of $6.25 \times 10^{-5}$. Despite the drop suffered at the beginning of the equilibrium training, SHINE in its refined version is able to perform on par with the Jacobian-Free method (Fung et al., 2021). We also confirm the importance of choosing the right initialization to perform accelerated backpropagation, by showing that with a limited iterative inversion, the performance of the original method deteriorates. Finally, while the drop in performance for the accelerated methods is significant when applied in their vanilla version, we remind the reader that no fine-tuning was performed on the training hyperparameters, making those results encouraging (on par with architectures like ResNet-18 (He et al., 2016)).
The key take-away from Figure 3 is that both SHINE and Jacobian-Free approximation methods allow to accelerate the DEQ's backward pass at a relatively low accuracy cost.[6] Moreover, using the proposed refined versions of these methods, the performance drop can be traded-off for acceleration.

## 4    CONCLUSION AND DISCUSSION

We introduced SHINE, a method that leverages the qN matrices from the forward pass to obtain an approximation of the gradient of the loss function, thereby reducing the time needed to compute this gradient. We showed that this method can be used on a wide range of applications going from bi-level optimization to small and large scale computer vision tasks. We found that both SHINE and the Jacobian-Free method reduce the required amount of time for the backward pass of implicit models, potentially lowering the barriers for training implicit models.
As those methods still suffer from a small performance drop, there is room for further improvement. In particular, a potential experimentation avenue would be to understand how to balance the efforts of the Adjoint Broyden method in order to come closer to guaranteeing the asymptotical correctness of the approximate inversion. On the theoretical side, this may involve the rate of convergence of the approximated gradient. It also seems desirable to develop a version of Theorem 4 in which convergence of $(z_n)$ to $z^\star$ is not an assumption but rather follows from the assumptions, as achieved in Theorem 3. We have no doubt that the contraction assumption used for the Jacobian-Free method would allow to prove such a result, but expect that a significantly weaker assumption will suffice.

---

[6]More on the overall computational effort can be found in Table E.2

## REPRODUCIBILITY STATEMENT

We provide with the submission of this paper the full code necessary to reproduce the figures and the other quantitative results of the paper, from the training, to the evaluation and the actual figure drawing. We made sure to use seeds and verified that the seeding was indeed allowing reproducible results. We also provide time estimates for the reproduction of the figures. We made sure to provide the full proofs for our theorems in the supplementary material of this manuscript. The core concepts used in the proofs, and their sketches are also laid out in the main text.

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

## A  OPA ALGORITHM

---

**Algorithm LBFGS:** (Limited memory) BFGS method with OPA

---

**Input:**  initial guess $(z_0, B_0^{-1})$, where $B_0^{-1}$ is symmetric and positive definite, tolerance $\epsilon > 0$,
 frequency of additional updates $M \in \mathbb{N}$, memory limit $L \in \mathbb{N} \cup \{\infty\}$, $(t_n)$ a null
 sequence of positive numbers with $\sum_n t_n < \infty$

Let $F := \nabla_z g_\theta$
**for** $n = 0, 1, 2, \ldots$ **do**
  **if** $\|F(z_n)\| \leq \epsilon$ **then** let $z^\star := z_n$ and let $B := B_n$; STOP
  Let $\hat{B}_n^{-1} := B_n^{-1}$
  **if** $(n \bmod M) = 0$ **then**
    let $e_n := t_n B_n^{-1} \frac{\partial g_\theta}{\partial \theta}\Big|_{z_n}$, $\hat{y}_n := F(z_n + e_n) - F(z_n)$ and $\hat{r}_n := (e_n)^T \hat{y}_n$
    **if** $\hat{r}_n > 0$ **then**
      let $\hat{a}_n := e_n - B_n^{-1} \hat{y}_n$ and let

$$\hat{B}_n^{-1} := B_n^{-1} + \frac{\hat{a}_n (e_n)^T + e_n (\hat{a}_n)^T}{\hat{r}_n} - \frac{(\hat{a}_n)^T \hat{y}_n}{(\hat{r}_n)^2} e_n (e_n)^T$$

  Let $B_n^{-1} := \hat{B}_n^{-1}$
  **if** $n \geq L$ **then** remove update $n - L$ from $B_n^{-1}$
  Let $p_n := -B_n^{-1} F(z_n)$
  Obtain $\alpha_n$ via line-search and let $s_n := \alpha_n p_n$
  Let $z_{n+1} := z_n + s_n$, $y_n := F(z_{n+1}) - F(z_n)$ and $r_n := (s_n)^T y_n$
  **if** $r_n > 0$ **then**
    let $a_n := s_n - B_n^{-1} y_n$ and let

$$B_{n+1}^{-1} := B_n^{-1} + \frac{a_n (s_n)^T + s_n (a_n)^T}{r_n} - \frac{(a_n)^T y_n}{(r_n)^2} s_n (s_n)^T$$

  **else** let $B_{n+1}^{-1} := B_n^{-1}$
  **if** $n \geq L$ **then** remove update $n - L$ from $B_{n+1}^{-1}$
**Output:** $z^\star$, $B$

---

**Remark.** *A possible choice for $(t_n)$ is to use an arbitrary $t_0 > 0$ and $t_n := \|s_{n-1}\|$ for $n \geq 1$.*

## B  PROOFS OF SHINE CONVERGENCE

To facilitate reading, we restate the results before proving them.

### B.1  CONVERGENCE USING ULI

**Theorem 2** (Convergence of SHINE to the Hypergradient using ULI). *Let us denote $p_\theta^{(n)}$, the SHINE direction for iterate $n$ in Algorithm 1 with $b =$ true. Under Assumptions 1 and 2, for a given parameter $\theta$, $(z_n)$ converges q-superlinearly to $z^\star$ and*

$$\lim_{n \to \infty} p_\theta^{(n)} = \frac{\partial \mathcal{L}}{\partial \theta}\Big|_{z^\star}.$$

*Proof.* Under Assumptions 1 and 2, More and Trangenstein (1976, Theorem 5.7) shows that $B_n$ satisfies

$$\lim_{n \to \infty} B_n = J_{g_\theta}(z^\star)$$

The inversion operator is continuous in the space of invertible matrices, so we have:

$$\lim_{n \to \infty} B_n^{-1} = J_{g_\theta}(z^\star)^{-1}$$

Because $\nabla_z \mathcal{L}$ and $\frac{\partial g_\theta}{\partial \theta}$ are continuous at $z^\star$ by Assumption 2 (iii), we also have thanks to Assumption 2 (i):

$$\lim_{n \to \infty} \nabla_z \mathcal{L}(z_n) = \nabla_z \mathcal{L}(z^\star) \qquad \text{and} \qquad \lim_{n \to \infty} \frac{\partial g_\theta}{\partial \theta}\Big|_{z_n} = \frac{\partial g_\theta}{\partial \theta}\Big|_{z^\star}$$

By continuity we then deduce that, as claimed,

$$\lim_{n\to\infty} p_\theta^{(n)} = \lim_{n\to\infty} \nabla_z \mathcal{L}(z_n) B_n^{-1} \frac{\partial g_\theta}{\partial \theta}(z_n) = \nabla_z \mathcal{L}(z^\star) J_{g_\theta}(z^\star)^{-1} \frac{\partial g_\theta}{\partial \theta}\Big|_{z^\star} = \frac{\partial \mathcal{L}}{\partial \theta}\Big|_{z^\star} \qquad \square$$

### B.2 CONVERGENCE FOR BFGS WITH OPA

**Assumption 5** (Extended Assumptions for BFGS). *Let $g_\theta(z) = \nabla_z r_\theta(z)$ for some $C^2$ function $r_\theta : \mathbb{R}^d \to \mathbb{R}$. Consider Algorithm 1 with $b = \mathtt{false}$ and suppose that*

1. *the set $\Omega := \{z \in \mathbb{R}^d : r_\theta(z) \leq r_\theta(z_0)\}$ is convex;*
2. *$r_\theta$ is strongly convex in an open superset of $\Omega$ (this implies that $r_\theta$ has a unique global minimizer $z^\star$) and has a Lipschitz continuous Hessian near $z^\star$;*
3. *there are positive constants $\eta_1, \eta_2$ such that the line search used in the algorithm ensures that for each $n \geq 0$ either*

$$r_\theta(z_{n+1}) \leq r_\theta(z_n) - \eta_1 \left[\frac{\nabla r_\theta(z_n)^T p_n}{\|p_n\|}\right]^2 \quad or \quad r_\theta(z_{n+1}) \leq r_\theta(z_n) + \eta_2 \nabla r_\theta(z_n)^T p_n$$

   *is satisfied;*
4. *the line search has the property that $\alpha_n = 1$ will be used if both*

$$\frac{\|(B_n - J_{g_\theta}(z_n))s_n\|}{\|s_n\|} \quad and \quad \|z_n - z^\star\|$$

   *are sufficiently small.*

**Remark.** *The requirements 3. and 4. on the line search are, for instance, satisfied under the well-known Wolfe conditions, see Byrd et al. (1988, section 3) for further comments.*

**Theorem 3** (Convergence of SHINE to the Hypergradient for BFGS with OPA). *Let us consider $p_\theta^{(n)}$, the SHINE direction for iterate $n$ in Algorithm 1 that is enriched by extra updates in the direction $e_n$ defined in (5). Under Assumptions 2 (ii-iii) and 3, for a given parameter $\theta$, we have the following: Algorithm 1, for any symmetric and positive definite matrix $B_0$, generates a sequence $(z_n)$ that converges q-superlinearly to $z^\star$, and there holds*

$$\lim_{n\to\infty} p_\theta^{(n)} = \frac{\partial \mathcal{L}}{\partial \theta}\Big|_{z^\star}. \tag{6}$$

*Proof.* The proof is divided into four steps. The first step is to establish the q-superlinear convergence of $(z_n)$ to $z^\star$. Denoting by $N_e \subset \{0, M, 2M, \ldots\}$ the set of indices of extra updates that are actually applied, the second step consists of showing

$$\lim_{N_e \ni n\to\infty} (B_n - J_{g_\theta}(z^\star)) \frac{e_n}{\|e_n\|} = 0, \tag{9}$$

where, in this proof, $B_n$ always represents the matrix from Algorithm LBFGS *before* the update in the direction $e_n$ is applied, i.e., the matrix whose inverse appears in the definition of $e_n$, while $\hat{B}_n$ always represents the matrix from Algorithm LBFGS *after* the update in the direction $e_n$ has been applied; if the update in the direction $e_n$ is not applied, then $B_n = \hat{B}_n$. The third step is to prove that (9) implies the desired convergence (6) of the SHINE direction if the limit $n \to \infty$ is replaced by $N_e \ni n \to \infty$, i.e., the limit is taken on the subsequence corresponding to $N_e$. The fourth step is then to transfer the convergence to the entire sequence.

It is easy to check that instead of updating $B_n^{-1}$, respectively, $\hat{B}_n^{-1}$, we can also obtain the sequences $(B_n)$ and $(\hat{B}_n)$ by updating according to

$$B_{n+1} = B_n + \frac{y_n y_n^T}{y_n^T s_n} - \frac{B_n s_n (B_n s_n)^T}{s_n^T B_n s_n}$$

for the usual update (skipping the update if $y_n^T s_n \leq 0$), respectively,

$$\hat{B}_n = B_n + \frac{\hat{y}_n \hat{y}_n^T}{\hat{y}_n^T e_n} - \frac{B_n e_n (B_n e_n)^T}{e_n^T B_n e_n}$$

for the extra update (skipping the update if $\hat{y}_n^T e_n \leq 0$). Here, the quantities $y_n$, $\hat{y}_n$ and $e_n$ are defined as in Algorithm LBFGS. We can now argue essentially as in the proof of Byrd et al. (1988,

Theorem 3.1) to show that $(z_n)$ converges q-superlinearly to $z^\star$. As part of that proof we obtain that $\hat{B}_n \neq B_n$ for at least $\lceil 0.5Q \rceil$ of the indices $n = 0, M, 2M, \ldots, QM$ for any $Q \in \mathbb{N}$ (namely for all $n \in N_e$ satisfying $n \leq QM$) and that we can apply Byrd and Nocedal (1989, Theorem 3.2), which yields

$$\lim_{n\to\infty} \left( \hat{B}_n - J_{g_\theta}(z^\star) \right) \frac{s_n}{\|s_n\|} = 0 \qquad \text{and} \qquad \lim_{N_e \ni n\to\infty} \left( B_n - J_{g_\theta}(z^\star) \right) \frac{e_n}{\|e_n\|} = 0. \qquad (10)$$

For the third step, we abbreviate $v_n := \frac{\partial g_\theta}{\partial \theta}|_{z_n}$. From the definition of $e_n$ and (10) we infer that

$$0 = \lim_{N_e \ni n\to\infty} \left( B_n - J_{g_\theta}(z^\star) \right) \frac{e_n}{\|e_n\|} = \lim_{N_e \ni n\to\infty} \left( I - J_{g_\theta}(z^\star) B_n^{-1} \right) \frac{v_n}{\|B_n^{-1} v_n\|}.$$

After multiplication with $J_{g_\theta}(z^\star)^{-1}$ this entails

$$\lim_{N_e \ni n\to\infty} \left( J_{g_\theta}(z^\star)^{-1} - B_n^{-1} \right) \frac{v_n}{\|B_n^{-1} v_n\|} = 0,$$

which shows that

$$\lim_{N_e \ni n\to\infty} B_n^{-1} v_n = \lim_{N_e \ni n\to\infty} J_{g_\theta}(z^\star)^{-1} v_n = J_{g_\theta}(z^\star)^{-1} \frac{\partial g_\theta}{\partial \theta}|_{z^\star}$$

by Assumption 2 (iii). Using Assumption 2 (iii) again it follows that

$$\lim_{N_e \ni n\to\infty} p_\theta^{(n)} = \lim_{N_e \ni n\to\infty} \nabla_z \mathcal{L}(z_n) B_n^{-1} \frac{\partial g_\theta}{\partial \theta}\Big|_{z_n} = \nabla_z \mathcal{L}(z^\star) J_{g_\theta}(z^\star)^{-1} \frac{\partial g_\theta}{\partial \theta}\Big|_{z^\star} = \frac{\partial \mathcal{L}}{\partial \theta}\Big|_{z^\star},$$

concluding the third step. To infer that (6) holds, it suffices to show that $\lim_{N_e \ni n\to\infty} \|B_n - B_{j_n}\| = 0$ for any sequence $(j_n)_{n \in N_e} \subset \mathbb{N}$ such that $\{j_n, j_n+1, \ldots, n-1\} \cap N_e = \emptyset$ for all $n \in N_e$ sufficiently large. Indeed, since for $C := \max\{\sup_n \|B_n\|, \sup_n \|B_n^{-1}\|\}$, which is finite by Byrd and Nocedal (1989, Theorem 3.2), there holds

$$(B_n) \subset \left\{ A \in \mathbb{R}^{d\times d} : A^{-1} \text{ exists }, \|A\| \leq C, \|A^{-1}\| \leq C \right\}$$

and the set on the right-hand side of the inclusion is compact by the Banach lemma, inversion is a *uniformly* continuous operation on this set, hence $\lim_{N_e \ni n\to\infty} \|B_n^{-1} - B_{j_n}^{-1}\| = 0$, so

$$\lim_{N_e \ni n\to\infty} \|p_\theta^{(n)} - p_\theta^{(j_n)}\| = 0$$

by continuity, and therefore

$$\lim_{N_e \ni n\to\infty} p_\theta^{(j_n)} = \lim_{N_e \ni n\to\infty} p_\theta^{(n)} = \frac{\partial \mathcal{L}}{\partial \theta}\Big|_{z^\star}$$

by the third step, establishing the claim.

It remains to show the validity of $\lim_{N_e \ni n\to\infty} \|B_n - B_{j_n}\| = 0$ for any sequence $(j_n)_{n \in N_e}$ such that $\{j_n, j_n + 1, \ldots, n - 1\} \cap N_e = \emptyset$ for all $n \in N_e$ sufficiently large. Since at least every second extra update is actually carried out, the condition on the intersection implies $n - j_n \leq 2M - 1$ for all these $n$. Now let $(j_n)_{n \in N_e}$ be any such sequence. Then $B_n - B_{j_n} = \sum_{m=j_n}^{n-1} B_{m+1} - B_m$ is a sum of at most $2M - 1$ BFGS updates in search directions, but contains no extra updates. Hence, the secant conditions $B_{n-l} s_{n-1-l} = y_{n-1-l}$, $l \in \{0, 1, \ldots, n - j_n\}$, are satisfied, allowing us to deduce

$$\|B_{n-l} - B_{n-l-1}\| = \frac{\|(B_{n-l} - B_{n-l-1}) s_{n-l-1}\|}{\|s_{n-l-1}\|}$$

$$\leq \frac{\|y_{n-l-1} - J_{g_\theta}(z^\star) s_{n-l-1}\|}{\|s_{n-l-1}\|} + \frac{\|(B_{n-l-1} - J_{g_\theta}(z^\star)) s_{n-l-1}\|}{\|s_{n-l-1}\|}$$

for all $l \in \{0, 1, \ldots, n - j_n - 1\}$. For each of these $l$, both terms on the right-hand side tend to zero for $N_e \ni n \to \infty$ (for the second term this follows from the first identity in (10) due to $B_{n-l-1} = \hat{B}_{n-l-1}$). Recalling that $B_n - B_{j_n} = \sum_{m=j_n}^{n-1} B_{m+1} - B_m$ we find $\lim_{N_e \ni n\to\infty} \|B_n - B_{j_n}\| = 0$, which finishes the fourth step and thus concludes the proof. $\qquad \square$

### B.3 Convergence for Adjoint Broyden with OPA

**Theorem 4** (Convergence of SHINE to the Hypergradient for Adjoint Broyden with OPA). *Let us consider* $p_\theta^{(n)}$, *the SHINE direction for iterate $n$ in Algorithm 1 with the Adjoint Broyden secant condition (7) and extra update in the direction $v_n$ defined in (8). Under Assumptions 2 and 4, for a given parameter $\theta$, we have q-superlinear convergence of $(z_n)$ to $z^\star$ and*

$$\lim_{n\to\infty} p_\theta^{(n)} = \frac{\partial \mathcal{L}}{\partial \theta}\Big|_{z^\star}.$$

*Proof.* Due to Assumption 2, the superlinear convergence of $(z_n)$ follows from Schlenkrich et al. (2010, Theorem 2). The proof of the remaining claim is divided into two cases.

Case 1: Suppose that $\nabla_z\mathcal{L}(z^\star) = 0$. By continuity this implies $\lim_{n\to\infty} \nabla_z\mathcal{L}(z_n) = 0$. Since the sequence $(B_n^{-1}\frac{\partial g_\theta}{\partial \theta}|_{z_n})$ is bounded by Assumption 4, it follows that

$$\lim_{n\to\infty} p_\theta^{(n)} = \lim_{n\to\infty} \nabla_z\mathcal{L}(z_n)B_n^{-1}\frac{\partial g_\theta}{\partial \theta}\Big|_{z_n} = 0 = \frac{\partial \mathcal{L}}{\partial \theta}\Big|_{z^\star},$$

as claimed.

Case 2: Suppose that $\nabla_z\mathcal{L}(z^\star) \neq 0$. By continuity this implies $\nabla_z\mathcal{L}(z_n) \neq 0$ for all sufficiently large $n \in \mathbb{N}$. Let us denote by $N_e \subset \mathbb{N}$ the set of indices of extra updates. We stress that this set is infinite since, by construction, every $M$-th update is an extra update. We have $v_n \neq 0$ for all sufficiently large $n \in N_e$, hence Schlenkrich et al. (2010, Lemma 3) yields

$$\lim_{N_e\ni n\to\infty} \frac{\|\nabla_z\mathcal{L}(z_n)(I - B_n^{-1}J_{g_\theta}(z^\star))\|}{\|(\nabla_z\mathcal{L}(z_n)B_n^{-1})^T\|} = \lim_{N_e\ni n\to\infty} \frac{\|(v_n)^T(B_n - J_{g_\theta}(z^\star))\|}{\|v_n\|} = 0.$$

This implies

$$\lim_{N_e\ni n\to\infty} \frac{\|\nabla_z\mathcal{L}(z_n)(J_{g_\theta}(z^\star)^{-1} - B_n^{-1})\|}{\|\nabla_z\mathcal{L}(z_n)B_n^{-1}\|} = 0,$$

thus necessarily

$$\lim_{N_e\ni n\to\infty} \|\nabla_z\mathcal{L}(z_n)(J_{g_\theta}(z^\star)^{-1} - B_n^{-1})\| = 0.$$

Since $\lim_{N_e\ni n\to\infty} \nabla_z\mathcal{L}(z_n)J_{g_\theta}(z^\star)^{-1} = \nabla_z\mathcal{L}(z^\star)J_{g_\theta}(z^\star)^{-1}$ by continuity, we find

$$\lim_{N_e\ni n\to\infty} \nabla_z\mathcal{L}(z_n)B_n^{-1} = \nabla_z\mathcal{L}(z^\star)J_{g_\theta}(z^\star)^{-1},$$

whence

$$\lim_{N_e\ni n\to\infty} p_\theta^{(n)} = \lim_{N_e\ni n\to\infty} \nabla_z\mathcal{L}(z_n)B_n^{-1}\frac{\partial g_\theta}{\partial \theta}\Big|_{z_n} = \nabla_z\mathcal{L}(z^\star)J_{g_\theta}(z^\star)^{-1}\frac{\partial g_\theta}{\partial \theta}\Big|_{z^\star} = \frac{\partial \mathcal{L}}{\partial \theta}\Big|_{z^\star}, \quad (11)$$

where we have used continuity again. To prove that these limits hold not only for $N_e \ni n \to \infty$ but in fact for all $\mathbb{N} \ni n \to \infty$, we establish, as intermediate claim, that for any fixed $m \in \mathbb{N}$ we have $\lim_{n\to\infty} \|B_{n+m} - B_n\| = 0$. Note that this claim is equivalent to $\lim_{n\to\infty} \|B_{n+1} - B_n\| = 0$. Denoting by $L \geq 0$ the Lipschitz constant of $J_{g_\theta}$ near $z^\star$, we find

$$\|B_{n+1} - B_n\| = \frac{\|v_n v_n^T [J_{g_\theta}(z_{n+1}) - B_n]\|}{\|v_n\|^2} \leq \|J_{g_\theta}(z_{n+1}) - J_{g_\theta}(z^\star)\| + \frac{\|[J_{g_\theta}(z^\star) - B_n]^T v_n\|}{\|v_n\|}$$

$$\leq L\|z_{n+1} - z^\star\| + \frac{\|E_n^T v_n\|}{\|v_n\|}.$$

Both terms on the right-hand side go to zero as $n$ goes to infinity: the first one due to $\lim_{n\to\infty} z_n = z^\star$ and the second one since $\lim_{n\to\infty} \frac{\|E_n^T v_n\|}{\|v_n\|} = 0$ by Schlenkrich et al. (2010, Lemma 3). This shows that $\lim_{n\to\infty} \|B_{n+1} - B_n\| = 0$, which concludes the proof of the intermediate claim.

From $\lim_{n\to\infty} \|B_{n+m} - B_n\| = 0$ for any fixed $m \in \mathbb{N}$ it follows that for any sequence $(j_n) \subset \mathbb{N}$ with $\sup_n |j_n - n| < \infty$ there holds $\lim_{n\to\infty} \|B_{j_n} - B_n\| = 0$. This implies for any such sequence $(j_n)$ the limit $\lim_{n\to\infty} \|B_{j_n}^{-1} - B_n^{-1}\| = 0$. To establish this, note that for $C := \max\{\sup_n \|B_n\|, \sup_n \|B_n^{-1}\|\}$, which is finite by Assumption 4 and the combination of the bounded deterioration principle (Schlenkrich et al., 2010, Lemma 2) with Assumption 2 (i), the set

$$\left\{ A \in \mathbb{R}^{d\times d} : A^{-1} \text{ exists}, \|A\| \leq C, \|A^{-1}\| \leq C \right\}$$

includes the sequence $(B_n)$ and is compact by the Banach lemma, so inversion is a *uniformly continuous* operation on this set.

Now let us construct a sequence $(j_n) \subset N_e$ by defining, for every $n \in \mathbb{N}$, $j_n := \arg\min_{m \in N_e} |n - m|$. That is, for every $n$, $j_n$ denotes the member of $N_e$ with the smallest distance to $n$. It is clear that $|n - j_n| \le M - 1$ for all $n$, hence $\lim_{n \to \infty} \|B_{j_n}^{-1} - B_n^{-1}\| = 0$. Using this and, again, continuity it is easy to see that

$$\lim_{n \to \infty} \|p_\theta^{(n)} - p_\theta^{(j_n)}\| = 0,$$

which implies by (11) that

$$\lim_{n \to \infty} p_\theta^{(n)} = \lim_{n \to \infty} p_\theta^{(j_n)} = \lim_{N_e \ni n \to \infty} p_\theta^{(n)} = \left. \frac{\partial \mathcal{L}}{\partial \theta} \right|_{z^\star},$$

thereby establishing the claim. $\qquad\qquad\qquad\qquad\qquad\qquad\qquad\qquad\qquad\qquad\square$

**Remark.** *An inspection of the proof reveals that if $B_n$ is never updated in the direction $z_n$, but only updated in the direction $v_n$ defined in (8), then Assumption 4 can be replaced by the significantly weaker assumption that the sequence $(B_n^{-1} \frac{\partial g_\theta}{\partial \theta}|_{z_n})$ is bounded. The price to pay is that the convergence rate of $(z_n)$ to $z^\star$ will be slower (q-linear instead of q-superlinear) since the updates in the direction $z_n$ are critical for ensuring fast convergence of $(z_n)$ to $z^\star$.*

## C    LOGISTIC REGRESSION HYPERPARAMETERS

For both datasets we split the data randomly (with a different seed for each run) between training-validation-test, with the following proportions: 90%-5%-5%. The hyperparameters are the same as in the original HOAG work (Pedregosa, 2016), except:

- We use a memory limitation of 30 updates (not grid-searched) for accelerated methods (Jacobian-Free and SHINE), compared to 10 for the original method. This is because the approximation should be better using more updates. We verified that using 30 updates for the original method does not improve the convergence speed. That number is 60 for OPA.

- We use a smaller exponential decrease of 0.78 (not grid-searched) for the accelerated methods, compared to 0.99 for the original method. This is because in the very long run, the approximation can cause oscillations.

We also use the same setting as Pedregosa (2016) for the Grid and Random Search. Finally, we highlight that warm restart is used for both the inner problem and the Hessian inversion in the direction of the gradient.

**OPA inversion experiments**    For the OPA experiments, we used a memory limitation of 60, and a tolerance of $10^{-6}$. The OPA update is done every 5 regular updates.

## D    DEQ TRAINING DETAILS

The training details are the same as the original Multiscale DEQ paper (Bai et al., 2020): all the hyperparameters are kept the same and not fine-tuned, and the data split is the same. We recall here some important aspects. For both datasets, the network is first trained in an unrolled weight-tied fashion for a few epochs in order to stabilize the training.

We also underline that the DEQ models, in addition to having a fixed-point-defining sub-network, also have a classification and a projection head.

Finally, for Figure 3, the median backward pass is computed with 100 samples on a single V100 GPU for a batch size of 32.

### D.1    CIFAR

The Adam optimizer (Kingma and Ba, 2015) is used with a $10^{-3}$ start learning rate, and a cosine annealing schedule.

### D.2    IMAGENET

The Stochastic Gradient Descent optimizer is used with a $5 \times 10^{-2}$ start learning rate, and a cosine annealing schedule.

The images are downsampled 2 times before being fed to the fixed-point defining sub-network.

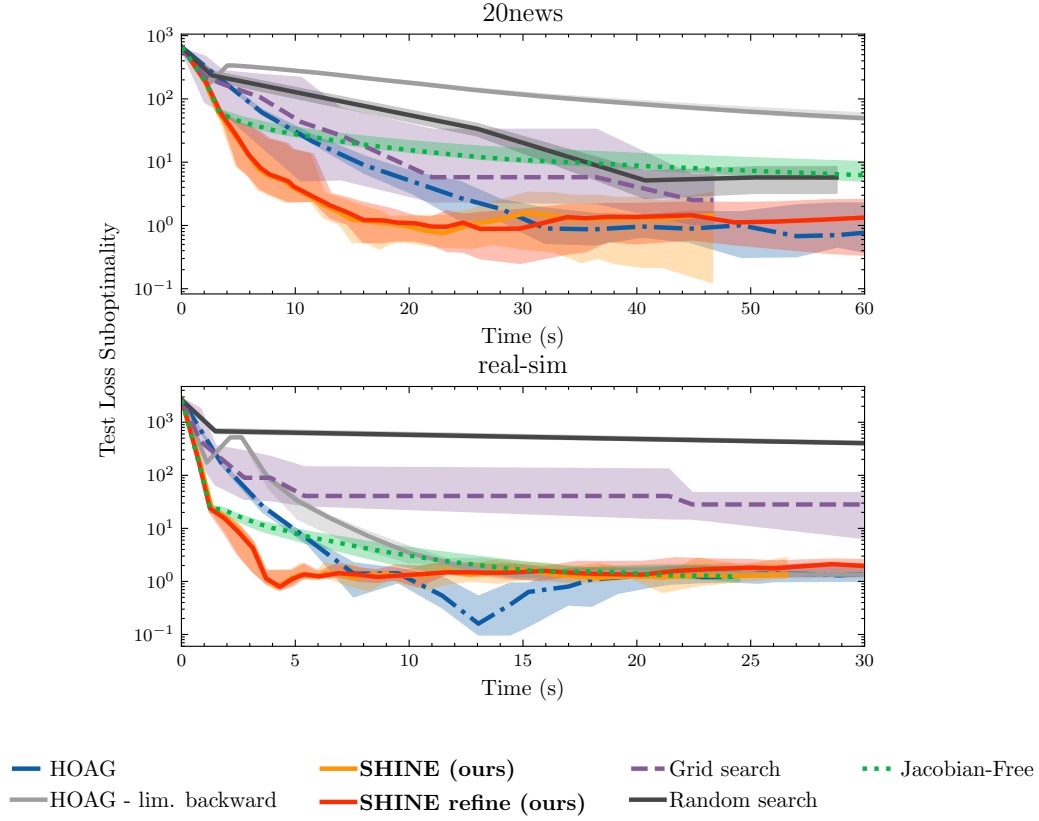

Figure E.1: **Bi-level optimization:** Convergence of different hyperparameter optimization methods on the $\ell_2$-regularized logistic regression problem for two datasets (20news (Lang, 1995) and real-sim (lib)) on held-out test data.

## E  ADDITIONAL RESULTS

### E.1  BI-LEVEL OPTIMIZATION EXTENDED

In order to make sure that SHINE was indeed improving over HOAG (Pedregosa, 2016), we also looked at the results obtained when performing an inversion with a precision lower than that prescribed by Pedregosa (2016) originally (i.e. truncating the iterative inversion). These results, also complemented with Random Search (Bergstra and Bengio, 2012), can be seen in Figure E.1. They confirm that the advantage provided by SHINE cannot be retrieved with a looser tolerance on the inversion.

### E.2  REGULARIZED NONLINEAR LEAST SQUARES

In order to further validate the efficiency of SHINE compared to competing methods, we also benchmarked it on the regularized nonlinear least squares task. For a training set $(x_{train,i}, y_{train,i})_{i=1}^N$ and a test set $(x_{test,i}, y_{test,i})_{i=1}^M$, this problem reads

$$\min_{\theta} \frac{1}{2} \sum_{i=1}^{M} \|y_{test,i} - \sigma((z^*)^\top x_{test,i})\|_2^2$$

$$z^* = \arg\min_{z} \frac{1}{2} \sum_{j=1}^{N} \|y_{train,j} - \sigma(z^\top x_{train,j})\|_2^2 + \frac{\theta}{2}\|z\|_2^2 \tag{12}$$

where $\sigma$ denotes the sigmoid function $\sigma(x) = \frac{1}{1+e^{-x}}$. For a fixed hyper-parameter $\theta$, this task is typically solved using L-BFGS (Berahas et al., 2021; Xu et al., 2020).

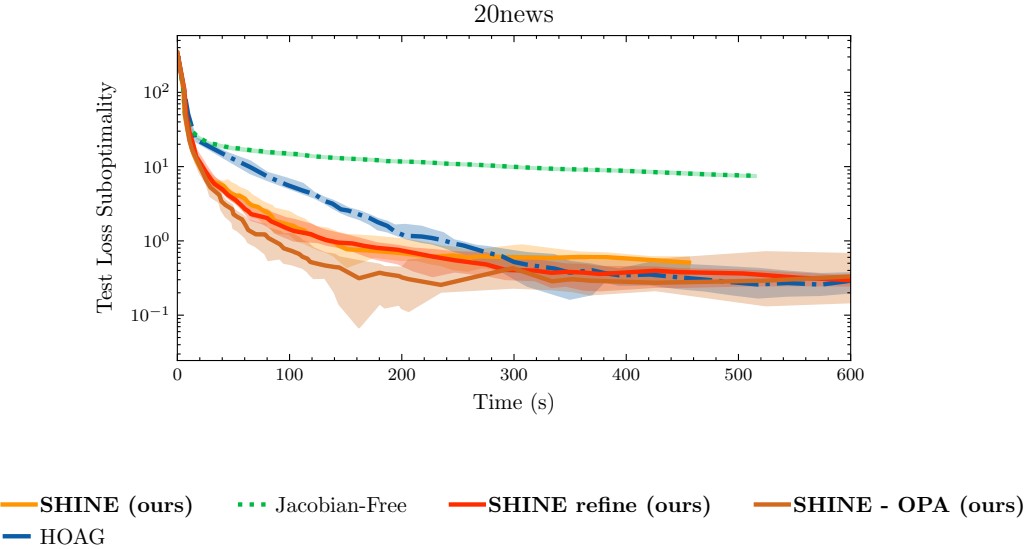

Figure E.2: **Bi-level optimization on regularized nonlinear least squares:** Convergence of different hyperparameter optimization methods on the $\ell_2$-regularized nonlinear least squares for the 20news (Lang, 1995) dataset on held-out test data.

Table E.1: Nonlinear spectral radius obtained by the power method for the fixed-point defining sub-network for the 3 different methods.

| Method | Nonlinear spectral radius |
|---|---|
| Original | 230.5 |
| Jacobian-Free | 193.7 |
| SHINE | 234.2 |

We can see in Figure E.2 that SHINE clearly outperforms the Jacobian-Free method and it is also quicker to converge compared to HOAG. We can also notice the benefit of OPA compared to the vanilla SHINE method is more pronounced. We hypothesize that this is due to the nonconvex nature of the inner problem making the Hessian inverse approximation more difficult, as was noted by Berahas et al. (2021).

### E.3 CONTRACTIVITY ASSUMPTION

One of the main limiting assumptions in the original Jacobian-Free method work (Fung et al., 2021), is the contractivity assumption. We showed here that it was not important to enforce this in order to achieve excellent results, but one can wonder whether this assumption is not met in practice thanks to the unrolled pretraining of DEQs. We looked at the contractivity of the fixed-point defining sub-network empirically by using the power-method applied to a nonlinear function, in the CIFAR setting. The results, summarized in Table E.1, show that the fixed-point defining sub-network is not contractive at all.

### E.4 TIME GAINS

Because the total training time is not only driven by backward pass but also by the forward pass and the evaluation, we show for completeness in Table E.2 the time gains for the different acceleration methods for the overall epoch. We do not report in this table the time taken for pre-training which is equivalent across all methods, and is not something on which SHINE has an impact. It is clear in Table E.2 that accelerated methods can have a significant impact on the training of DEQs because we see that half the time of the total pass is spent on the backward pass (more on ImageNet (Deng et al., 2009)). We also notice that while SHINE has a slightly slower backward pass than the Jacobian-Free method (Fung et al., 2021), the difference is negligible when compared to the total pass computational cost.

Table E.2: The time required for each method on the different datasets during the equilibrium training. For the forward and backward passes, the time is measured offline, for a single batch of 32 samples, with a single GPU, using the median to avoid outliers. This time is given in milliseconds. For the epochs, the time is measured by taking an average of the 6 first epochs, and given in hours-minutes for Imagenet and minutes-seconds for CIFAR. The epoch time for SHINE without improvement on Imagenet is not given because it never reaches the 26 forward steps: the implicit depth is too short. Fallback is not used for CIFAR. Numbers in parenthesis indicate the number of inversion steps for the refined versions.

| Dataset Name | CIFAR (Krizhevsky, 2009) | | | ImageNet (Deng et al., 2009) | | |
|---|---|---|---|---|---|---|
| Method Name | Forward | Backward | Epoch | Forward | Backward | Epoch |
| Original (Bai et al., 2020) | 256 | 210 | 4min40 | 644 | 798 | 3h38 |
| Jacobian-Free (Fung et al., 2021) | 249 | 12.9 | 3min10 | 621 | 13.5 | 2h02 |
| SHINE Fallback (ours) | 218 | 16.0 | 3min20 | 622 | 35.3 | 2h13 |
| SHINE Fallback refine (5, ours) | 272 | 96.6 | 3min50 | 622 | 212 | 2h44 |
| Jacobian-Free refine (5) | 260 | 86.5 | 3min40 | 620 | 186 | 2h43 |
| Original limited backprop | 281 | 86.4 | 3min50 | 653 | 187 | 2h40 |

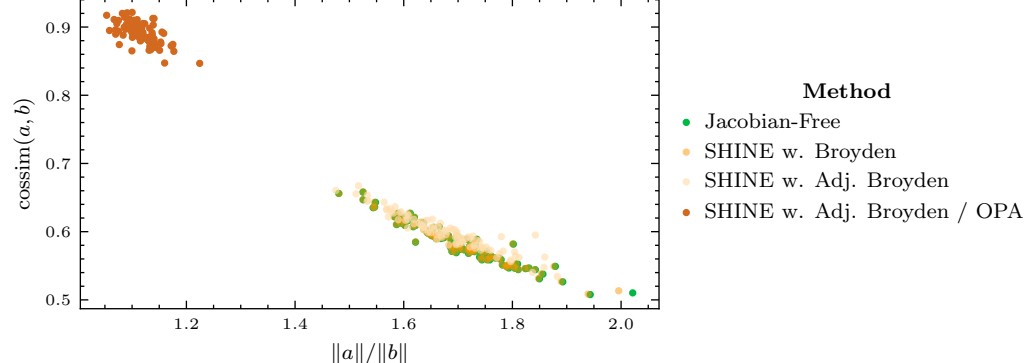

Figure E.3: **Quality of the inversion using OPA in DEQs :** Ratio of the inverse approximation over the exact inverse function of the cosine similarity between the inverse approximation $b = \nabla_z \mathcal{L}(z^\star) B_n^{-1}$ and the exact inverse $a = \nabla_z \mathcal{L}(z^\star) J_{g_\theta}(z^\star)^{-1}$ for different methods. For OPA, the extra update frequency is 5. 100 runs were performed with different batches.

### E.5 DEQ OPA RESULTS

We can clearly see in Figure E.3 that in the case of DEQs, OPA also significantly improves the inversion over the other accelerated methods. We also see that the improvements of SHINE over the Jacobian-Free method without OPA are marginal.

Because the inversion is so good, we would expect that the performance of SHINE with OPA would be on par with the original method's. However, this is not what we see in the results presented in Table E.3. Indeed, OPA does improve on SHINE with only Adjoint Broyden, but it does not outperform SHINE done with Broyden.

Table E.3: **CIFAR DEQ OPA results :** Top-1 accuracy of different methods on the CIFAR dataset, and epoch mean time.

| Methode name | Top-1 Accuracy (%) | Epoch mean time |
|---|---|---|
| Original | 93.51 | 4min40 |
| Jacobian-Free | 93.09 | 3min10 |
| SHINE (Broyden) | 93.14 | 3min20 |
| SHINE (Adj. Broyden) | 92.89 | 4min |
| SHINE (Adj. Broyden/OPA) | 93.04 | 4min40 |

