# OpenReview forum: "SHINE: SHaring the INverse Estimate from the forward pass for bi-level optimization and implicit models"
_ICLR.cc/2022/Conference — ICLR 2022 Spotlight_

### Official Review · Reviewer_H5f4 · 2021-11-01

**Correctness:** 4
**Technical Novelty And Significance:** 3
**Empirical Novelty And Significance:** 2
**Recommendation:** 8
**Confidence:** 3

**Main Review:**

Strong points:

The paper starts with a clear motivation and proposes an interesting approach to combine the computations from the forward and backward pass to accelerate the backward pass. The simplicity of their approach is an strength. Given the method, the paper provides theoretical analysis on the convergence of their forward and backward estimate to the desired computation. The theory is written well, seems correct, and states clear assumptions. The authors discuss the assumptions well (in case of whether they are used in any other works and if they are common). They provide detailed numerical results and experiments. In certain settings, they show faster performance than the original with similar performance.

Weak points:

Some of the experimental results are not convincing. For example, although they outperform Jacobian free method in the regularized logistic regression, they show similar performance to the Jacobian free in the case of classification with DEQ. So, why one may use SHINE which is slower than Jacobian free but have similar accuracy?

- An extensive experimental study is needed to compare the two (SHINE and Jacobian-free). For the proposed method to make practical sense, the author should show the wide settings at which SHINE is better than Jacobian free. For example, is SHINE better in any other bi-level optimization in addition to the regularized logistic regression?

- A discussion in the MAIN paper is needed for experimental studies on the relation between the quality of the inversion and the performance. I recommend to move Figure E.2 to the main text for this matter. Please explain why the quality of the inversion and performance are not correlated.

- Fallback strategy is of concern. Please provide more intuition of this instability and why it is not seen in other methods. How much is "barely" in the statement "we verified that the fallback is barely used?"

- The bullet points of contributions need to be more precise.

- Figure 1. Fix typo "Freee".


Post dicussion opinion: see the discussion. Given the additional HO experiment and elaborations in the abstract, I have revised my rating and I recommend acceptance of this paper.

**Summary Of The Paper:**

In implicit deep learning such as deep equilibrium models, computing the inverse Jacobian for the forward pass is computationally expensive. This paper propose an interesting approach to combine the information from the forward and backward pass to make an efficient estimate of the Jacobian inverse. In one approach, they propose to replace the Jacobian in the backward update with the quasi-Newton matrix, which is being already used/estimated in the forward pass solved by quasi-Newton method. Additionally, they propose an iterative update to the quasi-Newton matrix such that to helps its estimate toward the direction useful in the backward pass (they call this outer problem awareness).

They provide theoretical analysis of their proposed method and show that under certain conditions/assumptions, the forward pass still converges to the desired solution and the sequences of backward estimates converges to the loss gradient of needed to parameter updates. They provide numerical results in bi-level optimization (regularized logistic regression) and training DEQ for classification. In certain settings (bi-level optimization), they show that they outperform Jacobian-free and have similar performance to the state-of-the-arts but it is faster than all. For DEQ, they show similar performance to Jacobian-free.

**Summary Of The Review:**

The paper's main motivation is to computationally improve the backward pass. They show speed improvement compared to the other methods involving inversion of the Jacobian and minimal decrease in performance. However, they do not show performance improvement compared to the Jacobian free which is faster than the method in DEQ experiment. Given this, why one may SHINE that approximates the inverse Jacobian when their method does not outperform the Jacobian-free method? More experimental results is needed to highlight the advantage of their method against Jacobian-free. This advantage is already shown once in the regularized logistic regression.

---

> ### Author Response · Authors · 2021-11-15
> **Regarding the experimental validation of SHINE**
>
> We thank the reviewer for acknowledging that our idea is a natural one relevant for the bi-level community, that our theory is clear and well written and that we showed “faster performance” in the bi-level setting with detailed numerical results. We addressed the concern about the comparable performance between J-F and SHINE in **W1** and **W2**. The reviewer also states that  “extensive experimental study is needed” and we demonstrate below why we think our experimental setup already provides clear insights on our proposed method.
>
> Firstly, for hyper-parameter optimization, we reuse the experimental setting from the paper introducing the HOAG method (Pedregosa 2016). Up to our knowledge, this experiment is the canonical benchmark for bi-level methods with strongly convex inner function and L-BFGS. We demonstrate on this benchmark that our method clearly outperforms the Jacobian Free method and the others. These results corroborate that the proposed method works well in setups that match our theoretical results.
>
> Then, we turn to the larger scale, more complex task of training large scale multi scale DEQs. Our experiments are very large scale and quite extensive. We demonstrate that both Jacobian Free and our method are competitive to train such architecture and can help scale up the performances, and E.2 also corroborates our theoretical results.
>
> We now will try to tackle the rest of the points raised by the reviewer:
>
> - **Quality of the inversion and performance** We think that this discussion is interesting, but so far we do not have a satisfying answer as to why the performance is not correlated with the quality of the inversion for OPA. This is why we chose to leave this in the appendix.
> - **Fallback strategy** While we have tried hard to investigate this instability, we did not reach an understanding of where it came from. We will provide exact numbers to quantify the “barely”, which is indeed not precise enough.
> - **Contributions** We are not sure to which contributions the reviewer is referring. We would be glad to precise any one that seems unclear.
> - **Typo** We thank the reviewer for pointing it out and will correct it in the revision.
>
> While we agree that there are still some open directions to pursue and improve this line of work, we think this paper brings a valuable contribution to the community.

---

### Official Review · Reviewer_mXZ6 · 2021-11-03

**Correctness:** 3
**Technical Novelty And Significance:** 3
**Empirical Novelty And Significance:** 3
**Recommendation:** 8
**Confidence:** 3

**Main Review:**

Strengths:
- The theoretical aspects of the paper especially Theorems 3 and 4 are novel and good ideas.
- The theoretical part of the paper is written very well for someone unfamiliar with the literature.

Questions:
- Figure 1: On the 20 news dataset, the variance around the convergence of SHINE methods is a little concerning. Especially if one wishes to make the claim of "An acceptable level of performance is reached twice faster for the SHINE method compared to any other competitor".
- Again in Figure 2 left: The variance in the convergence curves of SHINE methods requires further comment by the authors. Perhaps it will be a good idea of having N runs and reporting average runtime improvement benefits over other methods along with a standard deviation.
- Figure 3: it seems that the Jacobian-free method and SHINE are almost equally performant in terms of top-1 accuracy while SHINE takes longer due to additional updates? While this is discussed briefly, I would like the authors to dig a bit deeper on why that is the case. As I understand right now, the authors claims are that the Jacobian free method is working outside the assumptions used to prove its convergence but it seems that even their method is well outside the scope of its assumptions? I am unsure what the claims are here. Why would one use SHINE over the Jacobian-free method for the DEQ models?


**Summary Of The Paper:**

The paper proposes a way to improve on the computational cost of bi-level optimization problems. These often come up in recently proposed Deep Equilibrium models and in hyperparameter optimization settings in ML.

**Summary Of The Review:**

Overall, I like the core idea and theoretical insights of the paper but have questions regarding the experiments as indicated in my main review. I am willing to update my scores based on author responses.

Update:

I would like to thank the authors for thoroughly engaging with reviewers on the platform. After reading the author responses to my review and other discussions on this forum, I am convinced that the improved draft should be presented at the conference and vote to accept the paper.

---

> ### Author Response · Authors · 2021-11-15
> **Regarding the high variance, speed of SHINE and the assumptions of accelerated methods**
>
> We thank the reviewer for appreciating the clarity of our manuscript as well as the novelty of our ideas and theoretical results and for their constructive feedback. The main weaknesses denoted by the reviewer are the variance in the hyper-parameter optimization experiments and the comparison with the Jacobian Free method (J-F). We have addressed the later in the common rebuttal (**W1**) and we thank you for noticing this high variance on both figures that we had not investigated as it was not in the original benchmark.
>
> - **Variance in Figure 1 and 2**: After careful investigation, the high variance is actually mainly due to our representation of the sub-optimality gap (the legend was not correct). The 1st and 9th quantile only represents a deviation of $1$ for a typical loss value of order $10^2$ (2 orders of magnitude below). For a better interpretation of these plots, we will include the median loss value in the captions. We will also change the quantiles from $0.1$ and $0.9$ to $0.25$ and $0.75$ in order to have a more robust assessment given we only use $N=10$ seeds controlling the data splits to run the experiments.
> - **Speed of SHINE w.r.t. Jacobian-Free (J-F)**: We partly answer this point in **W2**. We also highlight that in Figure 3., we use the same number of forward pass iterations for each method, and the same number of backward pass iterations for each data point of the same shape (as indicated in the legend).
> - **Assumptions on J-F**: Indeed, both methods are used outside the scope of their assumptions, but we did not mean this as a drawback of the J-F method but rather as a positive point. Indeed, one of our contributions is to provide strong confirmation of the usefulness of the J-F method in an extended setting compared to the original work.

---

> ### Author Response · Authors · 2021-11-25
> **Follow-up on our rebuttal after improvements to the paper's contributions statement and additional experiments**
>
> As the end of the discussion period is nearing, we would like to kindly ask reviewer mXZ6 for their feedback on our answers to their questions.
> First we would like to recall that the reviewer felt that the theoretical part of our work was “well-written” and “novel”, and that they were mostly questioning the experimental validation of our method.
> We think that the additional experiments we carried on the suggestion of reviewer H5f4 strengthen our results on Hyperparameter Optimization (HO). While the reviewer was questioning the variance, we think that this additional benchmark confirms that our results are solid. We also addressed the variance issue by using a more robust quantile estimation given our 10 runs.
> Regarding the questions on the Deep Equilibrium Models (DEQs), we further clarified our contributions in the abstract of our paper, to make it clear what exactly we added to the field.
> The end of the abstract now states:
> > We empirically study this approach and the recent Jacobian-Free method in different settings, ranging from hyperparameter optimization to large Multiscale DEQs~(MDEQs) applied to CIFAR and ImageNet. Both methods reduce significantly the computational cost of the backward pass. While SHINE has a clear advantage on hyperparameter optimization problems, both methods attain similar computational performances for larger scale problems such as MDEQs at the cost of a limited performance drop compared to the original models.
>
> As we feel we have answered the main points raised by the reviewer, we would appreciate it if the reviewer revised our paper’s rating accordingly.

---

### Official Review · Reviewer_tSim · 2021-11-04

**Correctness:** 4
**Technical Novelty And Significance:** 3
**Empirical Novelty And Significance:** 3
**Recommendation:** 8
**Confidence:** 3

**Main Review:**

I believe that this paper is studying a crucial problem. Both deep implicit/equilibrium models and hyperparameter optimization, which can be formulated as bilevel optimization problems, are important applications in machine learning. In particular, the computational bottleneck in hypergradient computation has long been a stumbling block for their applications in high dimensional settings. The proposed method of this paper borrows ideas from L-BFGS and Broyden’s method, which appears to be natural choices to consider. The authors establish various convergence results showing that the approximate hypergradients converge to the true hypergradients, under different sets of assumptions. Experimentally, the proposed method is comparable to or outperforms the Jacobian-Free method by Fung et al. (2021) in hyperparameter optimization for logistic regression and DEQs. The proposed method is interesting and can be viewed as a complementary method to the recent Jacobian-Free method.

Typos:
- Add punctuation whenever necessary in display style equations, like (1), Theorem 2, (4) and (6)
- page 8, Figure 2 caption: “wihtout” to “without”

**Summary Of The Paper:**

In various machine learning problems which can be formulated as a bilevel optimization problem and solved using gradient-based methods, the computation of hypergradients is necessary. However, the involved inverse Jacobian matrix in the hypergradient has been a computational bottleneck in high-dimensional settings. This paper proposes to use quasi-Newton matrices from the forward pass to approximate this inverse Jacobian matrix in the direction needed for the gradient computation which appears in the computation of hypergradients. The proposed algorithm is applied to both hyperparameter optimization and deep equilibrium models for CIFAR-10 and ImageNet, showing that it reduces the computational cost of the backward pass by up to two orders of magnitude.

**Summary Of The Review:**

This paper studies the problem of approximating the inverse Jacobian matrix in hypergradient computation for bilevel optimization problems solved with gradient-based methods, in an attempt to reduce the computational bottleneck of computing the exact inverse Jacobian matrix in high dimensions. The proposed method leverages quasi-Newton methods for such approximations. Experimental results have demonstrated the effectiveness of the proposed approach.

---

> ### Author Response · Authors · 2021-11-15
> **Thank you for your review**
>
> We thank the reviewer for their enthusiastic assessment of the paper and pointing out the typo in our manuscript. We will make sure to correct it in our revision. We will also update the punctuation in our equations.

---

### Author Response · Authors · 2021-11-15
**Regarding the performance of SHINE in the DEQs experiments**

We would like to thank the reviewers for their insightful comments which will help us improve the article. Most reviewers judged our work “*well written*”’, relevant and our theoretical results “*clear*”, yet pointing some concerns related to our experimental validation on DEQs and the comparison with the Jacobian-Free (J-F) method. From what we understand, all reviewers are convinced by our approach, our theoretical analysis and the results for HO experiments. These contributions alone constitute a strong addition to the community and we decided to go one step further, looking at DEQs, to broaden the impact of our work. (**W1**) The major weakness that is pointed out is that SHINE does not outperform the J-F method on large scale DEQ experiments. While we agree that this result is somewhat anticlimactic, not only do we not claim this anywhere in the paper, the demonstration of the effectiveness of J-F, a contemporary method, is also one of our contributions as well as its refined extension. Moreover, Figure E.2 demonstrates that our method is able to better estimate the hypergradient in the case of large DEQs, numerically validating our method. While exploring and understanding why this estimation improvement does not correlate with better performances for SHINE is of interest for future directions, we believe that our contributions are already significant.
(**W2**) We also believe that the speed advantage of J-F compared to SHINE is not significant when compared to the overall time for the couple forward-backward pass as shown in Table E.2, while the gain of both methods compared to the full inversion is. SHINE indeed takes indeed a slightly longer time to perform the backpropagation due to the tensor multiplication that needs to be carried out to apply the matrix $B^{-1}$.

We hope our answers will have resolved most of the issues raised in the reviews and that the reviewers will revise their initial rating in light of our comments.

---

> ### Comment · Reviewer_H5f4 · 2021-11-19
> **Comments on W1 and W2**
>
> I thank the authors for the response.
>
> Although I agree that showing J-F works well on large scale problems is new, this J-F result does not fit well with the main message of the paper. The paper aims to propose a new method on how the information can be shared within forward and backward pass to improve the inversion of the Jacobian with limited computational resources. Given this goal, the paper still misses a showcase with numerical results on DEQ to show the advantage of their method against J-F. Otherwise, with SHINE being slower than J-F, but with similar performance, why one may use SHINE. New experiments are needed indeed to convince the user to prefer SHINE over J-F. Hence, I do not find the reviewer's response addressing my following comments from earlier review:
>
> - For the proposed method to make practical sense, the author should show the wide settings at which SHINE is better than Jacobian free.
> - The quality of the inversion and its relation to the performance must be studied in more detail (Figure E.2).

---

> > ### Comment · Reviewer_tSim · 2021-11-19
> > **Comments on the experiments**
> >
> > As the only reviewer who champion the acceptance of the paper, I would like to make the following points:
> > - I find SHINE an interesting method proposed in this paper. I really wonder whether it has to outperform and be faster than other existing methods to make it a good contribution to the literature, even though most of the machine learning community are blindly chasing SOTA results nowadays. It is expected that SHINE should be slower than Jacobian-free methods in larger models like DEQs, but this doesn't mean that SHINE has no merits.
> >
> > - The performance of SHINE in HO experiments alone makes SHINE a valuable contribution, when taking into consideration the theoretical analysis of SHINE as well.
> >
> > - In general, users might choose to use Jacobian-free methods for DEQs. But the current results in this paper indicates that it is possible to borrow ideas from quasi-Newton methods to compute the hypergradients, which might eventually lead to more ground-breaking results in future effort.

---

### Author Response · Authors · 2021-11-29
**Thank you to all reviewers**

We want to take this last opportunity to write a message to thank all the reviewers for engaging in a fruitful, timely and respectful review process that led to an improved version of the paper.

---

### Decision · Program_Chairs · 2022-01-20

**Decision:**

Accept (Spotlight)

**Comment:**

The paper considers the setting of bi-level optimization and proposes a quasi-Newton scheme to reduce the cost of Jacobian inversion, which is the main bottleneck of bi-level optimization methods. The paper proves that the proposed scheme correctly estimates the true implicit gradient. The theoretical results are supported by numerical experiments, which are encouraging and show that the proposed method is either competitive with or outperforms the Jacobian Free method recently proposed in the literature.

Even though the reviews expressed some initial concerns regarding the empirical performance of the proposed method, the authors adequately addressed those concerns and provided additional experiments. Thus, a consensus was reached that the paper should be accepted.